# The Metabolome and the Gut Microbiota for the Prediction of Necrotizing Enterocolitis and Spontaneous Intestinal Perforation: A Systematic Review

**DOI:** 10.3390/nu14183859

**Published:** 2022-09-18

**Authors:** Laura Moschino, Giovanna Verlato, Miriam Duci, Maria Elena Cavicchiolo, Silvia Guiducci, Matteo Stocchero, Giuseppe Giordano, Francesco Fascetti Leon, Eugenio Baraldi

**Affiliations:** 1Neonatal Intensive Care Unit, Department of Women’s and Children’s Health, Padova University Hospital, 35128 Padova, Italy; 2Institute of Paediatric Research, Città della Speranza, Laboratory of Mass Spectrometry and Metabolomics, 35127 Padova, Italy; 3Paediatric Surgery, Department of Women’s and Children’s Health, Padova University Hospital, 35128 Padova, Italy; 4Laboratory of Mass Spectrometry and Metabolomics, Department of Women’s and Children’s Health, Padova University Hospital, 35128 Padova, Italy

**Keywords:** necrotizing enterocolitis, spontaneous intestinal perforation, metabolomics, microbiota, preterm birth

## Abstract

Necrotizing enterocolitis (NEC) is the most devastating gastrointestinal emergency in preterm neonates. Research on early predictive biomarkers is fundamental. This is a systematic review of studies applying untargeted metabolomics and gut microbiota analysis to evaluate the differences between neonates affected by NEC (Bell’s stage II or III), and/or by spontaneous intestinal perforation (SIP) versus healthy controls. Five studies applying metabolomics (43 cases, 95 preterm controls) and 20 applying gut microbiota analysis (254 cases, 651 preterm controls, 22 term controls) were selected. Metabolomic studies utilized NMR spectroscopy or mass spectrometry. An early urinary alanine/histidine ratio >4 showed good sensitivity and predictive value for NEC in one study. Samples collected in proximity to NEC diagnosis demonstrated variable pathways potentially related to NEC. In studies applying untargeted gut microbiota analysis, the sequencing of the V3–V4 or V3 to V5 regions of the 16S rRNA was the most used technique. At phylum level, NEC specimens were characterized by increased relative abundance of *Proteobacteria* compared to controls. At genus level, pre-NEC samples were characterized by a lack or decreased abundance of *Bifidobacterium*. Finally, at the species level *Bacteroides dorei, Clostridium perfringens* and *perfringens-like strains* dominated early NEC specimens, whereas *Clostridium butyricum, neonatale* and *Propionibacterium acnei* those at disease diagnosis. Six studies found a lower Shannon diversity index in cases than controls. A clear separation of cases from controls emerged based on UniFrac metrics in five out of seven studies. Importantly, no studies compared NEC versus SIP. Untargeted metabolomics and gut microbiota analysis are interrelated strategies to investigate NEC pathophysiology and identify potential biomarkers. Expression of quantitative measurements, data sharing via biorepositories and validation studies are fundamental to guarantee consistent comparison of results.

## 1. Introduction

Necrotizing enterocolitis (NEC) is the most devastating gastrointestinal emergency in preterm neonates, which still carries a high burden of neonatal morbidity and mortality worldwide. NEC is estimated to affect about 6% (3–9%) of very low birth weight infants (birth weight less or as 1500 g) [1,2,3], and its mortality rate can reach 50.9% in extremely low birth weight infants (<1000 g) with surgical disease [3]. NEC is related to several short- and long-term complications, chief among these short bowel syndrome, intestinal failure, and neurodevelopmental sequalae [3,4,5].

Although traditionally diagnosed on the basis of clinical and radiological findings according to Bell’s modified criteria [6,7], there is increasing acceptance that “NEC” may simply be one umbrella for a spectrum of multiple conditions with different pathophysiology leading to the same outcome of intestinal necrosis [8]. In particular, the lack of a reliable definition of NEC has been a relevant challenge for its understanding. Patel et al. have recently summarized criteria for NEC diagnosis according to different networks and collaborative groups, highlighting the importance of a global consensus on defining NEC to improve research on the topic and its outcomes [9].

Given its potential devastating consequences and the initial non-specific signs and symptoms [10,11], research on biomarkers for early prediction and diagnosis of NEC has flourished in the last decades [12,13]. The “omics” technologies allow a comprehensive and systematic detection of mediators for revealing mechanisms of diseases and host-pathogen interactions [14]. Among them, metabolomics enables to depict the ultimate phenotypic expression of the ongoing biochemical processes to a stimulus [13,15,16]. Similarly, microbiota analysis uses high-throughput sequencing technologies, such as 16S rRNA gene sequencing and shotgun metagenomics, to profile the genomic composition of a microbial community in a culture-independent manner [17,18].

In recent years, high-throughput metabolomics and gut microbiota profiling have already allowed neonatal research to advance our understanding in many aspects of preterm-related diseases and the role of several factors, such as nutrition, in shaping the health of premature infants.

Compared to the targeted technologies, which are biased by the present knowledge and by previous research’s findings, the untargeted approach enables the comprehensive and extensive investigation of molecules and pathogens in a biological fluid, without a-priori hypothesis. Metabolomics allows to explore a high number of low molecular weight metabolites (up to 5000 with mass spectrometry), whereas gut microbiota analysis permits the sensitive identification of uncultivable bacteria.

As a result, the correlations of these multi-omic clusters hold considerable promise for discovering new biomarkers of disease. Additionally, the integration of metagenomics and metabolomics information may advance our knowledge on the microbiota-host cross-talk, and may provide new functional insight into the role of nutrition, antibiotic strategies, and immunomodulation for neonatal health.

This systematic review aims to provide an updated perspective of the literature, in which untargeted metabolomics and gut microbiota analysis have been applied for the prediction and diagnosis of NEC.

## 2. Materials and Methods

### 2.1. Study Type

We conducted a systematic review according to the recommended “Preferred Reporting Items for Systematic Reviews and Meta-analyses” (PRISMA) guidelines [19]. The protocol for this systematic review is registered within PROSPERO (ID CRD42022302608).

### 2.2. Literature Search

The systematic database search (PubMed, MEDLINE Ovid, Scopus) was independently performed by three reviewers from inception to October 31 2021 using the following terms: “neonate” OR “neonates” OR “newborn” OR “newborns” OR “infant” OR “infants”, AND “metabolome” OR “metabolomics” OR “microbiota” OR “microbiome”, AND “necrotizing enterocolitis” OR “NEC” OR “spontaneous intestinal perforation” OR “SIP”. Additional articles were identified by a manual search of the cited references.

### 2.3. Inclusion Criteria

Titles were screened for relevance and duplications by three reviewers independently, with disagreements resolved by discussion. Cross-sectional studies and studies with “prospective-specimen-collection, retrospective-evaluation” (PRoBE) [20] (single-group or multiple-groups) were included if they evaluated neonatal metabolome and/or gut microbiota in preterm infants with NEC and/or SIP, compared to healthy controls. Only studies applying untargeted techniques and culture-independent molecular techniques were included. The definition of NEC according to Bell’s stage II or III was considered for qualitative analysis. Spontaneous intestinal perforation was defined as isolated intestinal perforation with the presence of free intraperitoneal air in the absence of pneumatosis intestinalis [21]. Whenever possible, results were assessed and interpreted according to the timing of samples’ collection: early-samples, collected in the first weeks of life and >72 h from NEC onset (NEC prediction); late-samples, collected within 72 h from NEC onset (NEC diagnosis). Case series, brief report, unpublished studies (e.g., conference abstracts), and animal studies, were excluded. Abstracts for which the full text was in a language other than English were excluded. In the case of unavailable full text/data, authors were contacted directly by the reviewers.

### 2.4. Outcomes

The primary objective was to determine the differences in metabolic and gut microbiota patterns between infants with NEC Bell’s stage II or III and healthy controls. As secondary outcomes we evaluated the differences in metabolic and gut microbiota patterns between infants with NEC and those with SIP, and between lethal NEC (NEC-related death) and NEC not associated with mortality.

### 2.5. Data Extraction and Synthesis

The results of studies applying metabolomic analysis were expressed as absolute or relative presence (% of subjects), fold change/fold change ratio (FC/FCR), area under the curve (AUC), relative intensity (RI) or peak intensity (PI), or arbitrary unit (AU), whenever possible. The results of studies applying untargeted gut microbiota analysis were expressed as absolute or relative presence (% of subjects), relative abundance (RA), reads, or odds ratio (OR), when these data were available. Indices of alpha diversity within samples through time as expressed by species richness (Chao-index or Chao-i), or by richness and evenness estimators (Simpson and Shannon diversity indices) [22,23] were collected, whenever mentioned. Beta-diversity indices, as measures of between samples diversity and phylogenetic distance, expressed by weighted and unweighted UniFrac metrics [24] were considered as well, when available.

## 3. Results

A total of 1191 studies were screened for full text extraction after the removal of 811 duplicates. Of the full texts evaluated, 1158 were excluded due to valid reasons (title and abstract screening, missing data, unavailable full-text, or for not including the intervention or outcome of interest, Figure 1). Thirty-three studies were selected. Six studies applied untargeted metabolomic analysis, 23 used untargeted microbiota analytic techniques, and four applied both (Table 1 and Table 2) [25,26,27,28,29,30,31,32,33,34,35,36,37,38,39,40,41,42,43,44,45,46,47,48,49,50,51,52,53,54,55,56,57]. Only five studies applying metabolomics and 20 studies evaluating gut microbiota were considered for qualitative analysis, as they applied the strict NEC definition of Bell’s stage II or III. Figure 1 shows the PRISMA flow diagram of the study. The PRISMA checklist is available as Appendix A. Table 3 [58,59,60,61,62,63,64,65,66,67,68] reports the characteristics of the excluded studies after eligibility evaluation.

### 3.1. Untargeted Metabolomic Analysis

Of 10 studies applying untargeted metabolomic techniques (Table 1) [25,26,27,28,29,30,31,32,33,34], four were considered for qualitative analysis as they defined NEC according to Bell’s stage ≥II [26,27,28,29]. An additional study was also included as it investigated any possible variability of results due to inclusion of stage I NEC cases [31], for a total of 43 NEC cases and 95 preterm controls. All studies were cross-sectional, with a prospective collection of samples and their retrospective analysis. Three studies utilized nuclear magnetic resonance (NMR) spectroscopy applied on urinary samples [26,27,31], while two studies applied ultra-performance liquid chromatography-mass spectrometry (UPLC-MS) to evaluate the faecal metabolomic profiles [28,29].

Two studies included early-collected samples (>72 h before NEC onset) [26,29], which were considered as potentially predictive of NEC. Morrow et al. [26] found no single metabolite associated with NEC cases in samples between 4 and 9 days of life, but reported a urinary alanine/histidine ratio >4 to have a sensitivity of 82% with a predictive value of 78% for NEC. The other studies did not find any difference in terms of metabolomic profiles between cases and controls.

Four studies included samples collected in proximity to NEC diagnosis (late-collected samples, within 72 h from NEC onset) [27,28,29,31]. Stewart et al. [29] showed five metabolites in stools of NEC cases having the highest variable importance plot score. These belonged to linoleate metabolism, C21-steroid hormone biosynthesis, leukotriene metabolism, and formation of prostaglandin from arachidonate. Rusconi et al. [28] found significant changes in the sphingolipid pathway between cases and controls, in particular with increased sphingomyelins and decreased ceramides in faecal samples of NEC cases before disease onset. This trend was confirmed by the targeted analysis and was characteristic of NEC Bell’s stage II and III.

In urine, Picaud et al. [27] reported significantly decreased relative intensity of N,N-dimethylglycine (N,N-DMG), betaine, creatinine and urea in three cases of late-onset NEC compared to controls. Similarly, Thoimadou et al. [31] identified 14 endogenous metabolites (mainly amino acids and organic acids) to be significantly lower in the urinary profiles of NEC neonates, with succinate, citrate, 4-hydroxybenzoate, proline, tyrosine, and fumarate, having the highest AUC in the ROC curves (0.89, 0.85, 086, 0.83, 0.80, and 0.82, respectively). Tyrosine, fumarate, and proline, were also discriminating metabolites in the targeted LC-MS/MS analysis. The authors identified a multi-biomarker based on tyrosine, arginine, and riboflavin, as having the highest diagnostic ability in discriminating NEC neonates from controls (AUC 0.963).

None of the studies evaluated the difference in metabolic profiles between cases affected by NEC and those affected by SIP.

### 3.2. Untargeted Metabolomic Analysis

Twenty-seven studies applying untargeted gut microbiota strategies were identified (Table 2) [25,26,29,32,35,36,37,38,39,40,41,42,43,44,45,46,47,48,49,50,51,52,53,54,55,56,57], of which 20 focused on NEC Bell’s stage ≥II and were considered for the qualitative analysis (254 NEC cases, 651 preterm infants, 22 term infants) [26,29,35,39,40,41,42,43,44,45,46,47,49,50,51,53,54,55,56,57]. The sequencing of the V3, V4, or V5 regions (alone or combined) of the 16S rRNA was the technique used most frequently [26,29,39,40,41,43,44,47,49,50,51,56,57].

At the phylum level, five main taxa appeared to be significantly different in NEC Bell’s stage ≥II cases compared to controls. *Proteobacteria* showed a higher RA in NEC cases compared to controls (61–90.7% vs. 19–56.4%), from 2 weeks before NEC [53] up to near or at disease onset [43,45,54]. In controls, Lindberg et al. [43] showed a significant shift towards a reduction in *Proteobacteria* abundance from the early time in the hospital course to later time points (corresponding to NEC diagnosis). Four studies, instead, did not reveal any significant difference in *Proteobacteria* abundance between the two groups [35,39,42,46].

Near or at NEC diagnosis, NEC Bell’s stage ≥II cases were characterized by a significant lower relative abundance of the phyla *Firmicutes* [43,54] (RA 9.1–28.1% vs. 55.9–57.8%), *Bacteroidetes* [45,54] (0–0.5% vs. 2.5–8.1%), and *Fusobacteria* [54], compared to controls. Nevertheless, Morrow et al. [26] demonstrated an opposite relationship with lower RA of *Proteobacteria* and higher RA of *Firmicutes* in cases early in life (4–9 DOL), and Mai et al. [45] also found predominance of *Firmicutes* in NEC at the earlier time point.

Contrasting results emerged regarding the phylum Actinobacteria, with two studies showing significantly lower levels in NEC cases [45,46] (RA 0.2–1.3% vs. 1.7–3.8%), and two others [43,53] demonstrating the opposite (3–6.8% vs. 0.4–2.9%).

At the class level, included studies showed a wider variety of results. In two studies [50,57], *Clostridia* were more represented in NEC cases, whereas in the other two [46,56] this class, alone or combined with *Negativicutes*, was expressed in lower proportion compared to healthy controls. Nevertheless, in a multivariate analysis, Sim et al. [50] found that the reads for *Clostridium* operational taxonomic unit (OTU) had an OR of 6.2 (85%CI, 1.8–21.7, *p* = 0.004) in discriminating NEC Bell’s stage II and III from controls.

Contrasting results emerged in relation to the abundance of *Gammaproteobacteria*, as well, with two studies reporting an increased proportion of this class in cases at the time of NEC onset [54,56], whereas another highlighted the opposite at two weeks of life [57].

No conclusions could be deduced at order and family levels.

At the genus level, NEC cases appeared to have higher levels of Bacillus [44] at an early post-partum stage, and of *Pasteurella* at 2 weeks of life [57]. Controls were characterized by higher abundances of the genera *Staphylococcus* [40], *Propionibacterium* [26], and *Enterococcus* [43,44], in this period of life, instead. In pre-NEC samples, Stewart et al. [29] demonstrated that a preterm gut community type characterized by high RA of *Bifidobacterium* could not be found in any sample from NEC patients. Similarly, Torrazza et al. [53] showed significantly lower numbers of *Bifidobacteria* in cases than in controls during the weeks before diagnosis. In near NEC onset, affected infants showed a slight increase in *Clostridium sensu strictu* [49], *Enterobacter* [51,57], *Staphylococcus* [51], *E. Coli* [55,57], and *Shigella* [57], and less *Veillonella* [55]. *Streptococcus* [46,51], *Enterococcus* [41,47,51], and *Veillonella* [43,46], were greater in healthy subjects. In general, studies which described the progression of gut microbiota through different timepoints demonstrated specific trends rather than single putative pathogens before NEC onset. Interestingly, Zhou et al. [57] revealed different changes according to the timing of NEC development (early vs. late).

Finally, at the species level, higher abundances of *Clostridium perfringens* and *perfringens-like strains* [40,41], *Bacteroides dorei* [40], and an OTU matching closest to *Klebsiella pneumoniae* [53], characterized NEC infants at early timepoints after birth. These subjects showed lower abundance of *Clostridium difficile* [40], instead. At around 4 weeks of life, Leach et al. [42] found *Corynebacterium striatum* and *Morganella morganii* to be more abundant in NEC samples than in those of controls. Additionally, these authors showed in general a trend towards more opportunistic pathogenic bacteria (like *Pseudomonas aeruginosas* and *Corynebacterium amycolatum*) in NEC cases. At NEC diagnosis, cases were dominated by *Clostridium butyricum* [35,41,49], *neonatale* [49], and by *Propionibacterium acnei* [39], whereas controls had more *Streptococcus salivarius* [39]. The main findings of the included studies are summarized in Table 2 and Figure 2.

No studies compared the stool microbiota of NEC infants to those of preterm infants affected by SIP and controls.

Among the strains which appeared to be related to lethal NEC were Bacteroidaceae in meconium [40] and *E. Coli* [55], in particular uropathogenic *E. Coli* (UPEC) (OR 4.1, *p* = 0.003) independently by the timing of NEC onset. Three studies demonstrated a negative association between lactate-producing bacilli [40], *Klebsiella* spp. [55], and *Clostridia* (*Veillonella*) [46], with mortality, instead.

As regards alpha diversity within samples through time as expressed by species richness (Chao-1 index), four studies did not find any difference between cases and controls [26,39,45,53], whereas one study reported lower Chao-i in NEC 1–5 days prior to symptoms [46], and in particular in those with lethal NEC. Concerning alpha diversity as expressed by richness and evenness estimators, the Simpson index did not differ between cases and controls in three studies in our review [26,40,43]. The Shannon diversity index, instead, appeared to be significantly lower in NEC Bell’s stage ≥II than controls, both early in life and at disease onset, as reported by six of the studies [35,44,46,54,56,57]. Two studies found no significant difference in the Shannon diversity index between the two groups [47,55]. One study found a reduced temporal development of this index in NEC samples, compared to an increase in controls [29].

Studies which performed principal coordinate analyses (PCoA) based on the weighted or unweighted UniFrac metrics showed a clear separation of NEC cases from controls early after birth [44], 1 or 2 weeks prior to NEC [45,53], and close to NEC diagnosis [44,45,46,54], indicating the presence of a certain phylogenetic distance between microbial communities in the two groups. Feng et al. [39] and Normann et al. [47], instead, did not find any clustering by case status.

### 3.3. Combined Untargeted Metabolomics and Gut Microbiota Analysis

Four studies [25,26,29,32] explored both metabolomic and gut microbiota profiles to highlight potential interrelationships between the patient’s metabolic pathways and the changes induced by gut microbial activity. However, only two studies [26,29] included patients with stage II or III according to modified Bell’s criteria. In the study by Morrow et al. [26], alanine was positively associated with early-onset NEC cases who were preceded by *Firmicutes* dysbiosis. Alanine was also directly correlated with the relative abundance of this phylum. Histidine, instead, was inversely associated with NEC cases preceded by *Proteobacteria dysbiosis* and had a strong inverse association with the interval between collection of the urine sample and case onset. The urinary alanine:histidine ratio was inversely associated with the relative abundance of *Propionibacterium*, which lacked in all NEC samples. Stewart et al. [29] grouped gut microbiota profiles into six clusters named preterm gut community types (PGCTs) as reflected by the dominant core OTUs. The sixth PGCT, dominated by *Bifidobacteria* and comprising exclusively healthy samples, was the one with the lowest abundance of metabolites associated with NEC.

## 4. Discussion

NEC remains one of the most common and yet inexplicable causes of death in preterm infants [69], therefore the search for its non-invasive prognostic and diagnostic biomarkers is one of the most intriguing topics in neonatology and has been summarized in several systematic reviews so far [12,70,71,72,73,74]. The multifactorial pathogenesis of this complex condition makes it challenging to unravel a single potential biomarker of the disease. The diagnosis of NEC still remains predominantly clinical, with few laboratory and radiological findings as surrogates to identify the severity. Rapid, bedside, point-of-care tests, to be performed prior to or at clinical manifestations, may help in guiding the management, for instance in the matter of feeding strategies, appropriate administration of antibiotics, or the need for urgent surgical intervention [14]. In fact, once the clinical and radiological picture are clearly evident, it may be too late to prevent disease progression and associated catastrophic outcomes [75].

With their hypothesis-free hypothesis-generating approach, the omics technologies may untangle a better understanding of the molecular processes responsible for NEC. The rise in the interest of omic approaches is demonstrated by the exponential increase in “omic” papers in the last 20 years [71,76]. Untargeted techniques applied to NEC have interrogated neonatal serum, urine, and stools, as easily collected biological samples and feasible to explore [10,13,14,71,75,77]. Metabolomics has several advantages over the other omics approaches by detecting the functional end points of cellular reactions and the direct result of a biochemical response to a stimulus [10,76]. As last downstream products of gene transcription and enzymatic pathways, metabolites provide a closer picture of the organism’s phenotype and its interaction with the environment. They can be virtually detected in any type of biological specimen and assayed by 1H-Nuclear Magnetic Resonance (1H-NMR) spectroscopy or mass spectrometry (MS). The former exploits the local magnetic field around a nucleus to elucidate the molecular structure of metabolites, whereas the latter separates metabolites using one of a variety of methods, including ultra-performance liquid chromatography (UPLC-MS) and gas chromatography (GC-MS) [72,76,77]. NMR spectroscopy has a relatively lower sensitivity and can detect a smaller number of metabolites compared to MS, but it needs minimal sample preparation, has a high reproducibility, and can be applied on tissue samples directly (for example intestinal tissue). MS, instead, owns a high sensitivity, proven an adequate sample preparation, detecting up to 5000 metabolites, but it has moderate reproducibility and requires a more demanding preparation of samples [78]. Additionally, it needs known standards to exactly identify metabolites of interest. These techniques are analytical distinct but complementary, allowing a wide and sensitive detection of functional low molecular weight metabolites [71]. While blood samples (serum) may better reflect the pathophysiological-metabolic changes of global events, such as NEC, they have the drawback of being invasively collected, as well as potentially associated with iatrogenic anemia [31]. Urine is a suitable non-invasive sample option, however, the accumulation of metabolic products depends on creatinine clearance, which varies according to gestational and postnatal age. Ideally, stool would provide a non-invasive mean which may better reflect the host-microbiota interaction in the gut lumen, the site of interest for NEC [75,77]. Nevertheless, faeces are usually not regularly produced by extremely preterm infants, especially in the first days of life, their composition is very complex and heterogeneous, rich in macromolecules and food-derived metabolites, making their analysis difficult with instrumental methods [75,79,80]. Interestingly, one study explored dried blood spots as a medium in NEC research [65]. However, this was not included in our review due to the application of a targeted approach to amino acids and acylcarnitines. Given the benefits and drawbacks of each biological fluid, it would be reasonable to explore urine, stools, and a small amount of plasma, or dried blood spots, in order to gather the majority of information and to avoid iatrogenic anemia.

Studies applying metabolomics in infants with NEC have been summarized by several reviews in the last 10 years [10,71,72,76,81]. As already highlighted by previous authors, there is a wide variability in populations’ inclusion criteria, timing of samples’ collection and type of analyzed biological fluids. In our review, only five metabolomic studies applied a rigorous definition of NEC. Although there does not appear to be a unifying metabolomic signature of NEC, some studies show interesting results with involvement of pathways related to inflammatory response [26,29], intestinal permeability [28], and energy depletion, potentially due to an inflammatory state [31].

Stewart et al. [29] were among the first to integrate sequence and metabolomic stool analysis in preterm neonates for NEC. They found five metabolites with a temporal increase prior to NEC diagnosis and to be discriminatory in the NEC samples at the time of diagnosis. These belonged predominantly to the linoleate metabolism, which may be involved in an inflammatory-mediated damage [81]. Another metabolite which may be related to inflammatory changes, lactate, was associated to late-onset NEC cases in Picaud et al. [27]. Intriguingly, the enzyme lactate dehydrogenase was overabundant in stools of infants prior to NEC development in the shotgun metagenomic analysis by Tarracchini et al. [82], supporting the accumulation of lactate in these subjects possibly due to an imbalance between lactate-producing and lactate-utilizing bacteria. Rusconi et al. [28] were the only group to validate their initial untargeted observations in a targeted analysis of samples on an expanded, not fully independent, cohort. They showed significantly increased sphingomyelins and decreased ceramides in pre-NEC stools of Bell’s stage II and III cases versus Bell’s stage I and controls. The authors attributed these changes to a decrease degradation of sphingolipids, as they were not related to dietary differences. These are bactericidal components of cell membranes against gram bacteria, and they have been hypothesized to be relevant to NEC development as they can alter microbiome composition [83].

Finally, Thoimadou et al. [31] highlighted differences in several metabolic pathways between NEC cases and controls, with the highest impact expressed by amino acids’ metabolism. The reduction in some amino acids in cases, namely alanine, asparagine, and proline, was hypothesized to promote the Krebs cycle in order to maintain enterocytes’ supply energy and integrity during an inflammatory state. The low urinary alanine level in NEC is apparently in contrast with the results by Morrow et al. [26], but the timing of sample collection differed between the two studies, and findings by Morrow et al. were strictly related to the microbiome dysbiosis dominated by *Firmicutes*. An early metabolic deviation involving predominantly amino acids has also been reported by Sinclair et al. [65], who found reduced levels of tyrosine and increased levels of its precursor, phenylalanine, in NEC, similarly to Thoimadou and colleagues [31]. Both authors explain this alteration with a potential impairment in the conversion of phenylalanine into tyrosine occurring in critically ill neonates during catabolic and oxidative stress states [31,65]. Interestingly, similarly to Rusconi et al. [28], Thoimadou and colleagues [31] did not demonstrate any possible influence of NEC stage I on the LC-MS data, consistent with the thinking that stage I NEC may be more similar to feeding intolerance than Stage II and III NEC.

The role of microbial dysbiosis prior to NEC is supported by the fact that NEC cannot be produced in germ free animals, as well as by the positive association between antibiotic use and the disease [84,85]. Animal studies suggest that microbiota composition in the neonatal period may affect gastrointestinal (GI) tract development, mucosal integrity, and even nutritional status [86,87]. Additionally, although NEC does not occur in utero, meconium is not sterile and its microbial composition, most likely reflecting the in-utero environment, varies depending on gestational age, with the potential involvement in the subsequent development of sepsis and NEC [40,81,88]. After birth, neonatal microbiota is influenced by mode of delivery, early nutrition, type of nutrition, as well as drug administration. Additionally, the lactation stage appears to play a significant role in the breast milk microbiota, thus affecting the intestinal microbiome composition of infants, too [89,90]. Finally, antibiotics may delay intestinal colonization of potentially beneficial bacteria and reduce the diversity of the microbiome, thus predisposing to NEC [84]. For these reasons, the investigation of the gut microbiota through culture-independent techniques has increased steadily in the last years. Ideally, the use of both culture-based and culture-independent approaches should be complementary, as the first allows the isolation of bacteria at low levels in samples and even when undetectable by quantitative PCR (qPCR), while the latter enables the identification of uncultivable bacteria [35,41]. The most common molecular untargeted strategy, the 16S rRNA amplicon sequencing and community profiling, is based on the amplification of the V2–V4 regions. This approach has the advantages of generating the lowest error rate when assigning taxonomy, and of providing more reliable richness estimation when converting longer reads of the V3-V4 regions [47]. Nevertheless, the drawback of culture-independent data is the inability to resolve strains with “pathogenic potential” from non-pathogenic members of a lineage [55], as well as the reliability on the PCR method without the potential of detecting plasmids, eukariotes, and viruses, like the genome-resolved methods do [48].

In our review, four studies suggested a predominance of *Proteobacteria* [43,45,53,54] and two studies a reduced abundance of *Firmicutes* [43,54] in cases near disease onset. Two studies [26,45], instead, reported a reverse trend in the first week of life, with a dramatic shift related to these two phyla over the week before NEC diagnosis (around 34–50% increase in *Proteobacteria* in NEC cases). These findings suggest a change in the gut microbiota occurring after birth and progressing towards NEC development. Preterm infants do have a dynamic pattern of early intestinal colonization [81,85], which in NEC patients seems to be characterized by gram-positive cocci at the beginning, then overtaken by gram-negative facultative anaerobic organisms, counterbalanced by a gradually increasing abundance of anaerobes [29,36,47,50,81]. This evidence has also been supported by Pammi et al. [91], who conducted a rigorous review and meta-analysis of eight studies applying stool microbiome profiling and revealed a consistent trend towards higher relative abundances of *Proteobacteria* and decreased relative abundances of *Firmicutes* and *Bacteroidetes* around 30 weeks of corrected GA in NEC patients. Additionally, studies exploring the microbiome in intestinal specimens showed that *Proteobacteria* were the most abundant phyla in NEC infants (49.0%) [66], and significantly higher in these patients compared to those affected by SIP [67]. An opposite trend seems to occur in healthy controls, with a progressive shift towards more gram-positive bacteria (*Firmicutes* and *Bacteroidetes* phyla, genera *Staphylococci* and lactate-producing bacilli), converging to an “adult-like” microbiota phenotype [18,92,93].

Despite the discrete uniformity of findings at the phyla level, data on bacteria abundances at class, genus, and species levels showed a wider heterogeneity of results. Among gram-positive bacteria, the genera *Enterococcus* and *Veillonella* (belonging to *Firmicutes*) [26,43,44,46,47,51,57], and *Propionibacterium* and *Bifidobacterium* (belonging to *Actinobacteria*) [29,41,53] were more abundant in healthy controls than in affected subjects prior to NEC. These strains have indeed been described as beneficial for a healthy intestinal development and function [94]. Additionally, in a recent review and network meta-analysis Van den Akker et al. [95] demonstrated the superiority of combinations of one or more *Bifidobacterium* spp. and one or more *Lactobacillus* spp. in probiotics’ composition in reducing the risk of severe NEC. The combination of *Enterococcus* spp, *Bacillus* spp, one or more *Bifidobacterium* spp and *Streptococcus salivarius* seemed to produce the largest reduction in NEC onset [96,97,98]. Among gram-negative bacteria, from our review three species emerged to be prevalent in NEC patients and all belonging to *Proteobacteria phylum*, in particular *Bacteroides dorei* [40], *Morganella morganii* [42] and a strain matching closest to *Klebsiella pneumoniae* [53]. The first is a known intestinal commensal strain which has been demonstrated to be involved in autoimmune diseases, like Type 1 diabetes. The latter two, instead, are renowned causes of nosocomial infections. The recent large-scale gut microbiome meta-analysis performed by Tarracchini et al. in 2021 [82] has also shown an increase in relative abundance of opportunistic pathogens in NEC, including members of the *Klebsiella* genus. Gram-negative bacteria are rich in lipopolysaccharide, which binds to toll-like receptor 4 (TLR-4), known to be involved in the inflammatory response of NEC [76]. The role of other gram-negative bacteria, like *Pseudomonas* and *Escherichia coli* genera, is still controversial in the included studies.

As regards anaerobes, there are several studies applying culture and non-culture methods that support the association of the genera *Clostridium* and *Clostridium sensu strictu* [49,57] with NEC, in particular with a predominance of the species *Cl. Butyricum* [35,41,49] and its close-related *Cl. Neonatale* [49,99]. The first species was first described in NEC several decades ago [100], whereas the latter was associated with a NEC outbreak in a NICU [101]. The more well-known *Clostridium perfringens* and *perfringens-like strains* were found in faecal samples both early in life and near NEC diagnosis in two studies in our review [40,41]. These species have been associated to NEC development, especially with a fulminant course, in previous researches and case reports [102,103,104]. Similarly, Tarracchini et al. found a predominance of *C. perfrigens* and *C. neonatale* in NEC compared to healthy preterms in the analysis of only pre-NEC samples [82]. These findings suggest that some early microbial signatures involving nosocomial pathogens, which may likely be a consequence of broad-spectrum antibiotics treatment, should be further investigated.

In our review, the majority of studies found no difference in the Chao-i [26,39,45,53], but a cauterization of samples according to case status from several weeks before NEC up to NEC diagnosis by UniFrac metrics [26,44,46,53,54]. As suggested by Torrazza et al. [53], this may indicate that patients with NEC may have a total number of bacteria similar to that of healthy controls, but a difference in the kinds of OTUs present and their proportions. While the Simpson index was also comparable between cases and controls, similar to the Chao-i [26,40,43], the Shannon diversity index, another measure accounting for both abundance and evenness of the species present in a community, was lower in NEC cases in six out of eight studies [35,44,46,54,56,57]. A limited Shannon diversity usually characterizes all preterm infants, with an average of less than 20 OTUs in contrast to more than 200 for human adults [93]. However, a much more reduced index in NEC patients may indicate a decreased uniformity of individual distribution among species, and not just a lower number of species. Furthermore, given the non-complex gut microbial populations of preterm infants, the overrepresentation or underrepresentation of single taxons make it difficult to ascribe a host phenotype to a variation in diversity [105]. Pammi et al. [91] reported no differences in the observed species, Shannon diversity, and Simpson diversity indices, between NEC cases and controls, nor did the authors found any clustering of samples based on weighted and unweighted UniFrac distance. The shotgun metagenomics data collection by Tarracchini et al. [82], instead, showed that cases with manifested NEC (64 samples) were characterized by a lower index of bacterial species richness on average than that of age-matched healthy preterms. All in all, a statistically significant reduction in biodiversity in NEC compared to preterm healthy controls according to gestational age and postnatal age still needs further evaluation.

We explored studies applying both metabolomic and microbiota analysis to illustrate potential connections of the human gut according to disease status. Morrow and colleagues [26] found an association between individual urine amino acids and the microbiome preceding NEC development. In this study, alanine, which is ubiquitously incorporated into bacterial cell wall biosynthesis, especially into peptidoglycan of most gram-positive organisms, was increased in NEC preceded by *Firmicutes* and inversely correlated with the relative abundance of *Proteobacteria* and *Propionibacterium*. The study of the metabolome-microbiome interplay [106,107], despite being increasingly popular, needs further investigation. Studies, like the MAGPIE [108], will hopefully shed a light on this linkage [71]. Studies investigating preclinical alterations in fecal volatile organic compounds (VOCs) also harbor great potential for this purpose, as these chemicals are considered to reflect gut microbiota composition and concurrent metabolic activity in the host [74]. Indeed, several studies have demonstrated that VOC profiles of infants with NEC can be discriminated from those of controls from 2 to 4 days before onset of symptoms [109,110,111].

Our review has different limitations. Firstly, as highlighted by many previous reviews, the majority of included studies comprises a small sample size, with variable inclusion criteria, uncontrolled confounding biases and different applied techniques, therefore hindering consistent comparison of data [10]. Secondly, none of the included studies compared NEC versus SIP, which is the most common surgical disease in the differential diagnosis of NEC. These two entities share a similar clinical presentation, although with some different features. Therefore, it would be a major interest to compare SIP vs. NEC cases at the time of diagnosis in order to highlight the different pathogenetic mechanisms and guide therapeutic management. Another limitation of our study is the lack of accurate and detailed metadata analysis due to missing exact quantitative parameters, such as peak values, sensitivity, specificity, predictive values, and areas under the curve from most of the studies. These quantitative data are often not specified in the methods and results sections. Comprehensible and comparable information for future science would be possible if studies reported exact mean values with error indicators for every significant parameter, as well as the degree of significance of elevation in the comparison of study groups. Finally, caution must be paid in data interpretation due to the numerous host variables that may affect the host’s metabolome and gut microbiota. In fact, as NEC progression usually occurs over 8–9 weeks in extremely preterm infants, several factors are implied in disease onset [28]. Given this consideration, the ideal time to predict NEC development would be the first week of life. However, samples collected at this time may not be as diriment as those collected closer to NEC development [10,60]. We suggest, whenever possible, to non-invasively collect the majority of samples, from birth onwards, as well as at suspicion of disease onset, in order to capture the metabolic and microbiome changes through time.

Although we did not perform a meta-analysis based on raw 16S rRNA gene sequence data, we believe this article clearly summarizes the current evidence and provides potential inspiration points for future research. In Figure 3**,** we provide a possible flowchart for future studies with the aim of improving uniformity of data collection and quantitative measures. Infants with SIP should be included as this entity goes in differential diagnosis with NEC, has a steady prevalence, and is a major reason for peritoneal drainage and abdominal surgery in ELBWI [112]. Validation cohorts with targeted approach and culture-based analysis are mandatory to corroborate and confirm previous “high-risk” putative biomarkers. It is probable that a sequential collection of samples rather than only one sample collected at a single timepoint would be relevant in the understanding of the host-pathogen interactions in NEC development. Algorithm comprising maternal and neonatal variables (clinical or radiological) and characteristic metabolic-microbiota patterns may be the key to early unravel surgical NEC [113,114,115].

## 5. Conclusions

Untargeted metabolomic and gut microbiota analyses applied to non-invasive specimens are valuable tools to understand NEC pathophysiology and identify its early biomarkers. These strategies are dampened by the influence of multiple confounding factors, and by the limited data on GA and age-specific reference values. The inclusion of bigger samples sizes, of patients affected by NEC or SIP as strictly defined by major guidelines, and of quantitative measures of results, is mandatory for future studies in order to improve uniformity of data. Data sharing via biorepositories is fundamental to guarantee consistent comparison of data. Future studies need to include validation cohorts to confirm the results.

## Figures and Tables

**Figure 1 nutrients-14-03859-f001:**
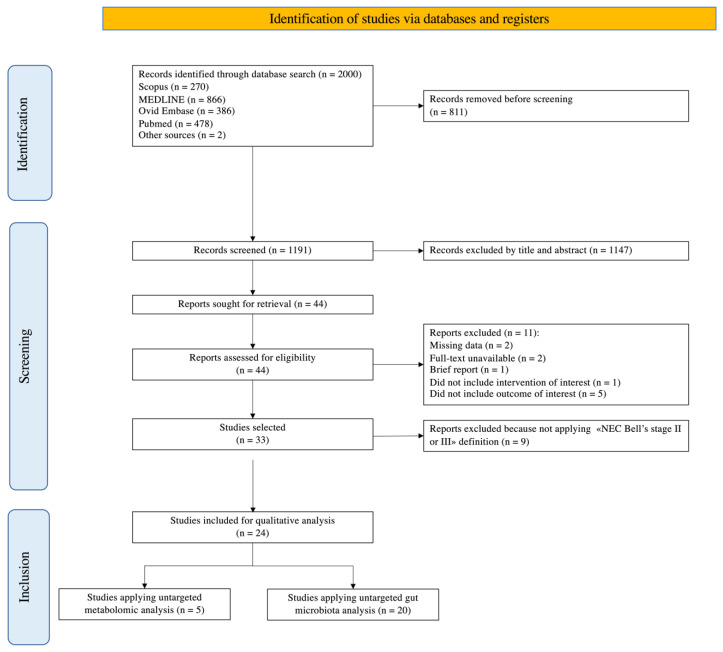
PRISMA flow diagram of the study.

**Figure 2 nutrients-14-03859-f002:**
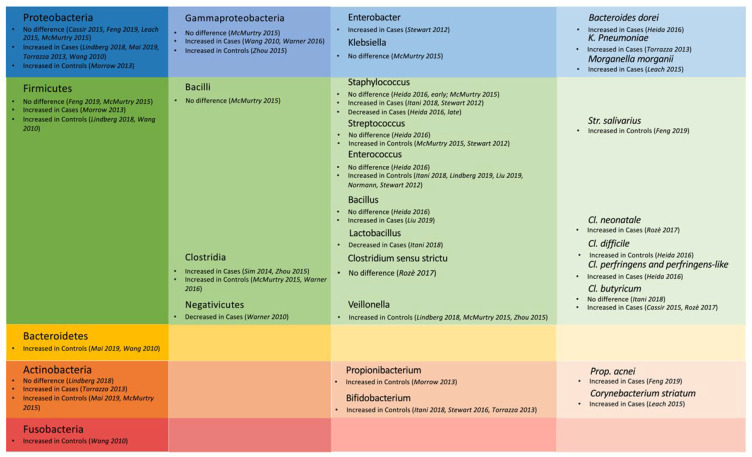
Main findings from the studies applying gut microbiota analysis at the phyla, classes, genera, and species level, who are predominant in cases and controls. References [26,29,35,39,40,41,42,43,44,45,46,47,49,50,51,53,54,56,57].

**Figure 3 nutrients-14-03859-f003:**
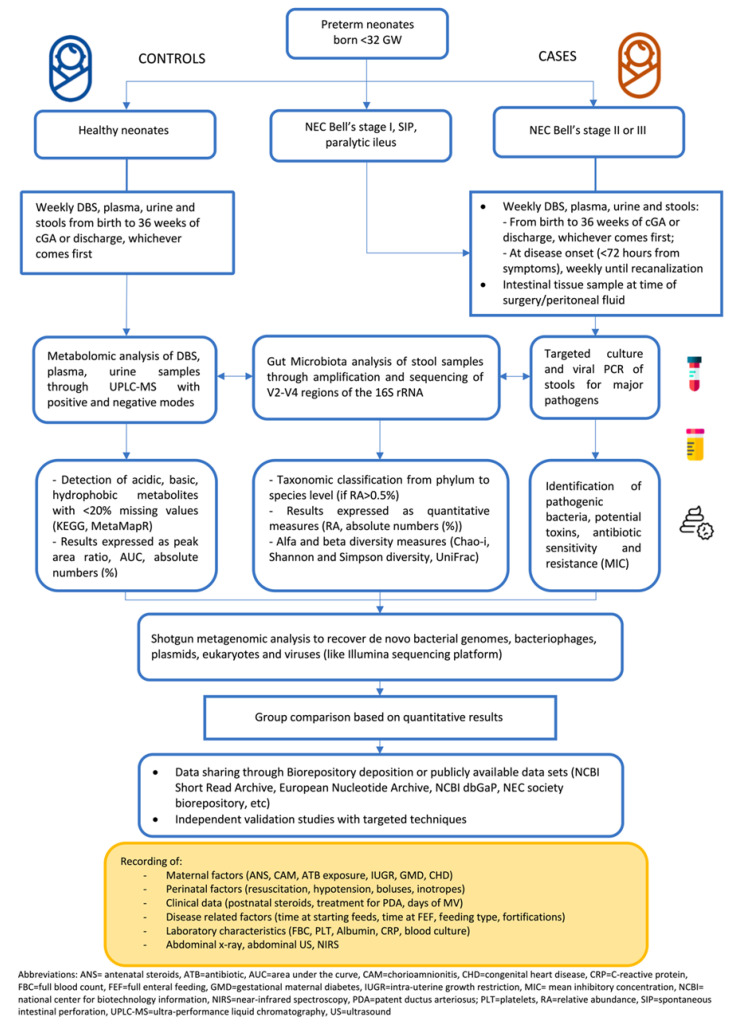
Proposed flow chart to conduct future studies applying untargeted metabolomics and microbiota analysis in the exploration of biomarkers for NEC.

**Table 1 nutrients-14-03859-t001:** Characteristics of studies applying untargeted metabolomic analysis for the predisposition/diagnosis of NEC/SIP (NEC/SIP prediction: early samples, first 14 days or >72 h from NEC onset; NEC diagnosis: late samples, <72 h from NEC onset). In grey shadow the studies which used NEC Bell’s stage II or III as definition and which were therefore included in the review. Numerical data are expressed as mean (SD) or median (IQR) or as number/percentage, if not otherwise specified.

Author/Year/Country	Study Design, Technique Applied	Inclusion Criteria (GA/BW) n NEC Definition	Mean (SD)/Median (IQR/Range) GA (Weeks)	Mean (SD)/Median (IQR) BW (Grams)	Sample Type and Timing	Increased Metabolites Cases vs. Controls (Number of Cases/Total Cases)	Decreased Metabolites Cases vs. Controls (Number of Controls/Total Controls)	Comments (Male Gender, C-Section, Antenatal Steroids, Antepartum Antibiotics, Enteral Feeding)
Brehin 2020 France [25]	Cross-sectional; NMR spectroscopy	<34 w Cases: 11 NEC Controls: 21 Suspected NEC by neonatologist (excluding CCC and SIP)	Cases: 28.4 (26–31) Controls: 30 (26.6–32)	Cases: 1150 (845–1815) Controls: 1360 (700–2105)	Stool every 10 days (1–10 d, 11–20 d, 21–30 d, >30 d)	None	Cases: lower Ethanol (>30 d); Serine (11–20 d); Leucine (>30 d) Controls: higher Serine (11–20 d); Ethanol (>30 d), Leucine (>30 d)	Cases: Male 9 (81.8%) CS NI ANS 11 (100%) Antepartum ATB 4 (36%) EBM NI Time of onset NI Controls: Male 13 (61.9%) CS NI ANS 19 (90%) Antepartum ATB 5 (24%) EBM NI
Morrow 2013 Ohio USA [26]	Cross-sectional; NMR spectroscopy	<29 w, <1200 g Cases: 11 NEC Controls: 21 NEC II or III Bell’s stage Subtypes of NEC based on ordination of day 4–9 samples: -NEC-I (*n* = 4) dominated by Firmicutes (Bacilli) -NEC-II (*n* = 5) dominated by Proteobacteria (Enterobacteriaceae)	Cases: 25.5 (1.8) Controls: 25.9 (1.5)	Cases: 791 (212) Controls: 839 (187)	Urine; T1 4–9 days; T2 10–16 days	T1: Increased Alanine and histidine normalized PI in NEC-I subtype compared to NEC-II subtype and controls; Cases: Alanine/histidine > 4 (9/11, 82%) Controls: A/H > 4 (5/20, 25%)	Controls: decreased alanine normalized PI;	Cases: Male 6 (54.6%) CS 7 (63.6%) ANS 10 (90.9%) Antepartum ATB 6 (54.6%) EBM (M or D) 11 (100%) Time of onset: NEC-I 7–21 d, NEC-II 19–39 d Controls: Male 8 (38.1%) CS 14 (66.7%) ANS 19 (90.5%) Antepartum ATB 10 (47.6%) EBM (M or D) 11 (100%)
Picaud 2021 France-Italy [27]	Cross-sectional; NMR spectroscopy	VLBW Cases: 6 NEC (3 early-onset, 3 late-onset) Controls: 12 (6 with feeding intolerance FI; 6 with good digestive tolerance GDT) NEC defined by clinical evidence fulfilling Bell’s stage criteria and with radiological pneumatosis intestinalis (stage ≥ II)	Cases: 27.1 (1.6) Controls 1: 27.2 (1.3) Controls 2: 27.7 (1.6)	Cases: 1016 (104) Controls 1: 920 (104) Controls 2: 950 (65)	Urine; Cases: before and at disease onset; Controls: at birth and as close as those of babies with NEC	Cases- Early-onset (<25 days): no differences; Cases-Late-onset (>40 days): increased lactate RI;	Controls GDT: increased N,N-DMG, betaine, myo-inositol, creatinine, urea RI;	Cases: Male 3 (50%) CS 2 (33.3%) EBM NI ANS, Antepartum ATB NI Time of onset < 25 days (3, 50%); >40 days (3, 50%) Controls 1: Male 3 (50%) CS 5 (83.3%) EBM NI ANS, Antepartum ATB NI Controls 2: Male 3 (50%) CS 3 (50%) EBM NI ANS, Antepartum ATB NI
Rusconi 2018 MO USA [28]	Prospective; UPLC-MS/MS for BRM, then targeted analysis of 14 ceramides and 7 sohingomyelins	Inclusion criteria NI Broad range metabolomics (BRM): Cases: 9 Controls: 19 Targeted: Cases: 23; Controls: 46 NEC Bell’s stage II and III, no SIP	BRM: Cases: 25.9 Controls: 25.1 Targeted: Cases: 25.9 (24.7–27.35) Controls: 25.5 (25–27.5)	BRM: Cases: 825.2 Controls: 787.3 Targeted: Cases: 800 (720–955) Controls: 840 (662.5–927.5)	BRM: stools closed to NEC onset (<5 days preceding it, not from the same day); Targeted: pre-event stool (1–3 days before NEC onset)	Cases (NEC II and III): increased sphingomyelins (not specified) Peak values expressed for the metabolites of the targeted analysis	Cases: low ceramides (not specified)	Targeted: Cases: Male 14 (61%) CS 19 (83.6%) EBM (M or D) 15 (65.2%) ANS NI, Antepartum ATB NI Time of onset 24 days Controls: Male 31 (67.2%) CS 29 (68.4%) EBM (M or D) 31 (67.4%) ANS NI Antepartum NI
Stewart Microbiome 2016 UK [29]	Prospective collection, retrospective analysis; UPLC-MS	<32 GW Cases: 7 NEC Controls: 28 Metabolomic analysis on 6 cases and 10 controls NEC “defined rigorously” by one senior clinician and two senior research clinicians, and classified as either surgical or medical, where pneumatosis was required for medical cases	Cases: 26 (23–30) Controls: 27 (24–30)	Cases: 760 (500–1470) Controls: 910 (545–1810)	Stools; (DOL) −14 (time point 1; TP1), −7 (TP2), 0 (TP3), +7 (TP4), and +14 (TP5)	TP3 (disease diagnosis): 5 metabolites with highest VIP scores: linoleate metabolism (2), C21-steroid hormone biosynthesis and linoleate metabolism (2), leukotriene metabolism and prostaglandin formation from arachidonate (1)		Cases: Male 3 (42.9%) CS 3 (42.9%) Time of onset 26.4 (14–42) days; ANS, EBM, Antepartum ATB NI Controls: Male 20 (71.4%) CS 15 (53.6%) ANS, EBM, Antepartum ATB NI
Stewart 2016 Ped Res UK [30]	Prospective collection, retrospective analysis; UPLC-MS	Inclusion criteria NI Cases: 10 (6 NEC, 4 LOS) Controls: 9 NEC categorized independently by attending physician and blindly confirmed	Cases NEC: 26.3 Controls: 26.2	Cases NEC: 922.5 Controls: 982.2	Serum; Cases: 14 days (+/−7) prior and 14 days (+/−4) after diagnosis	No unique metabolites characterizing patients with NEC compared to controls	No unique metabolites characterizing patients with NEC compared to controls	Cases: CS 2 (33.3%) EBM 5 (83.3%) Time of onset 24.2 days (16–31) Male, ANS, Antepartum ATB NI Controls: CS 4 (44.4%) EBM 8 (88.9%) Male, ANS, Antepartum ATB NI
Thomaidou 2019 Greece [31]	Prospective cross-sectional; untargeted NMR spectroscopy and targeted LC-MS	Preterm neonates (GA not specified); Cases: 15 (5 NEC I, 10 NEC II/III) Controls: 15 NEC every grade; separate sub-group analysis after excluding stage I NEC cases	Cases: 34 (29–36) Controls: 33.5 (29–36)	Cases: 2030 (1100–2680) Controls: 1815 (1130–2640)	Urine at time of evaluation for NEC; Cases: 8 (4–22) d Controls: 9 (4–34) d		Cases: Low Tyrosine (FC −1.6, AUC 0.80), Proline (FC −0.26, AUC 0.83), Citrate (FC −1.3, AUC 0.85), 4-hydroxybenzoate (FC −1.09, AUC 0.86), Formate (FC −1.33, AUC 0.82), Succinate (FC −1.0, AUC 0.89), 4-hydroxyphenylacetate (FC −1.23, AUC 0.78), Fumarate (FC −1.54, AUC 0.82), Creatinine (FC −0.35, AUC 0.79), Myo inositol (FC −0.24, AUC 0.79), hippuric acid (FC −1.46, AUC 0.76)	Cases: Male 9 (60%) CS 10 (66.7%) ANS 9 (60%) Antepartum ATB, EBM, Time of onset NI Controls: Male 8 (53.3%) CS 12 (80%) ANS 10 (66.6%) Antepartum ATB, EBM NI
Wandro 2018 California USA [32]	Retrospective; GC-MS	VLBW (<1500 g) Cases: 3 (NEC) Controls: 21 NEC definition NI	Cases 1: 25.6 Cases 2: 25.6 Controls: 27.4	Cases 1: 920 Cases 2: 776.4 (1 missing) Controls: 1018.6	Stools between days 7 and 75 of life (variable among patients)	No metabolites associated with NEC	No metabolites associated with NEC	Cases: Male NI CS 1 (33.3%) ANS, Antepartum ATB NI EBM 1 (33.3%) Mean time of onset 33 days Controls: Male NI CS 16 (76.2%) ANS, Antepartum ATB NI EBM 12 (57.1%)
Wang 2019 China [33]	Prospective collection, retrospective analysis; GC-MS	Cases (<35 GW, BW < 2.2 g): 19, 4 with NEC Controls (full-term): 20, 1 with NEC NEC Bell’s stage ≥1	Cases: 29.6–34.4 Controls: NI	Cases: 1500–2100 Controls: NI	Serum; before feeding	No differences related to NEC diagnosis but only to prematurity	NI	No data specified
Wilcock 2015 UK [34]	Prospective collection, retrospective analysis; GC-MS	Cases 1 (GA at birth <30 GW, NEC): 7 Cases 2 (GA at birth <30 GW): 5 Controls (GA at birth ≥37 GW): 8 NEC definition NI	Cases 1: 25 (24–28) Cases 2: 27 (25–29) Controls: 38.5 (37–40)	Cases 1: 799 (653–942) Cases 2: 987 (855–1209) Controls: 2983 (2642–3820)	Serum; T1: first 7 days of life; T2: At full enteral feeds (≥180 mL/kg/die)	No differences at T1; T2: 16 different metabolites (*p* < 0.05), 10/16 aminoacids with a FCR 0.26–8.19, high false discovery rate (only 3 samples from NEC cases); among these glycine, L-serine, decanoic acid, methionine, phenylalanine, ornithine and lysine (interleukin−1β pathway)	No differences at T1; T2: 16 different metabolites (*p* < 0.05), 10/16 aminoacids with a FCR 0.26–8.19, high false discovery rate (only 3 samples from NEC cases); among these glycine, L-serine, decanoic acid, methionine, phenylalanine, ornithine and lysine	Cases 1: Male 3 (42.9%) CS, ANS, Antepartum ATB NI EBM at full feeds 3 (60%) Time of onset 28 (15–35) days Cases 2: Male 4 (80%) CS, ANS, Antepartum ATB NI EBM at full feeds 2 (28.6%) Controls: Male 5 (62.5%) CS, ANS, Antepartum ATB NI EBM at full feeds 7 (87.5%)

Abbreviations: ANS = antenatal steroids; ATB = antibiotics; AU = arbitrary unit; AUC = area under the curve; BRM = broad range metabolomics; CAM = chorioamnionitis; CCC = complex congenital cardiopathy; CS = cesarean section; EBM = exclusive breast milk; FC = fold change; FCR = fold change ratio; F = French; FI = feeding intolerance; g = grams; GA = gestational age; GDT = good digestive tolerance; GW = gestational weeks; h = hours; MP = multiple pregnancy; NEC = necrotizing enterocolitis; NI = no information; NMR = nuclear magnetic resonance; N,N-DMG = N,N-Dimethylglycine; PI = peak intensity; RI = relative intensity; SIP = spontaneous intestinal perforation; TP = timepoint; VIP = variable importance plot; VLBW = very low birth weight; w = weeks.

**Table 2 nutrients-14-03859-t002:** Characteristics of studies applying untargeted gut microbiota analysis for the prediction/diagnosis of NEC/SIP (NEC/SIP prediction: early samples, first 14 days or >72 h from NEC onset; NEC diagnosis: late samples, <72 h from NEC onset)**.** In grey the studies defining NEC as Bell’s II or III and therefore included in the review. Numerical data are expressed as mean (SD) or median (IQR) or as number/percentage, if not otherwise specified. Not all pathogens reported in the seventh and eighth columns show a significant difference between cases and controls.

Author/Year/Country	Study Design, Technique Applied	Inclusion Criteria (GA/BW) n NEC Definition	Mean (SD)/Median (IQR) GA (Weeks)	Mean (SD)/Median (IQR) BW (Grams)	Sample Type and Timing	Increased Pathogens Cases Decreased Pathogens Cases	Increased Pathogens Controls Decreased Pathogens Controls	Diversity Index (Chao-i or Others)	Comments (Male Gender, C-Section, Antenatal Steroids, Antenatal ATB, Time at NEC Onset)
Brehin 2020 France [25]	Monocentric case-control; 16S bacterial DNA V3–V4 regions analysis by MiSeq	<34 w Cases: 11 NEC1 Controls: 21 Suspected NEC by neonatologist (excluding CCC and SIP)	Cases: 28.4 (26–31) Controls: 30 (26.6–32)	Cases: 1150 (845–1815) Controls: 1360 (700–2105)	Stool every 10 days (1–10 d, 11–20 d, 21–30 d, >30 d)	Increased Cases: *Streptococcus* spp., *Staphylococcus* spp., Micrococcales High intragroup variance	Increased Controls: *Klebsiella* spp. High intragroup variance	1–10 d Cases: lower Chao-i, *p* < 0.05; 11–20 d Cases: increased Chao-i, *p* < 0.05; No difference in Shannon and Simpson	Cases: Male 9 (81.8%) CS NI ANS 11 (100%) Antepartum ATB 4 (36%) EBM NI Time of onset NI Controls: Male 13 (61.9%) CS NI ANS 19 (90%) Antepartum ATB 5 (24%) EBM NI
Cassir 2015 France [35]	Prospective case-control; V6 region of 16S rRNA pyrosequencing to develop specific qPCR assay for Clostridium butyricum tested on the 2nd cohort	Preterm neonates (not specified) First analysis: Cases: 15 NEC (10 stage II, 5 stage III) Controls: 15 Second confirmation analysis: Cases: 93 Controls: 270 NEC defined as Bell’s stage II or III	Cases: 28.2 (2.7) Controls: 29 (2.8)	Cases: 1127 (380) Controls: 1220 (506.9)	Stools on the day of symptoms’ onset and on the same matched day for controls	Increased Cases: Clostridium butyricum (11/15, *p* < 0.01)	Decreased Controls: C. butyricum (2/15)	Shannon diversity index lower in NEC than controls (*p* = 0.035); OTUs decreased in NEC than controls (*p* < 0.0001)	Cases: Male 10 (66.7%) CS 11 (73.3%) ANS, Antepartum ATB, EBM NI Mean time of onset 18.2 (12.6) d Controls: Male 9 (60%) CS 12 (80%) ANS, Antepartum ATB, EBM NI
Claud 2013 IL USA [36]	Cross-sectional; V3–V4 region of 16S rRNA gene sequencing; Shotgun metagenomics-based analyses to examine gene function;	GA 24–32 GW Cases: 5 (NEC) Controls 1: 5 preterms Controls 2: 8 full-term breast-fed infants NEC definition not specified	Cases: 26.8 (24.4–32) Controls 1: 24.4 (24.1–32)	NI	Stools; weekly from birth to 10 weeks of life	Increased Cases: Proteobacteria Decreased Cases: Firmicutes Gene sets differentially abundant between two twins: the one with NEC with more carbohydrate metabolism	Increased Controls: Proteobacteria and Firmicutes	NI	Cases: Male 2 (40%) CS 5 (100%) ANS, Antepartum ATB NI EBM 2 (40%) Time of onset NI Controls 1: Male 3 (60%) CS 5 (100%) ANS, Antepartum ATB NI EBM 2 (40%)
Dobbler 2017 Brazil [37]	Cross-sectional; V4 region of 16S rRNA gene sequencing; Metagenomic sequencing	GA ≤ 32 GW Cases: 11 NEC Controls: 29 NEC diagnosed by neonatologist based on clinical criteria (abdominal distention, gastric aspirates, bilious vomiting, bloody stools, lethargy, apnea, hypoperfusion); in 9 patients pneumatosis intestinalis, in 2 patients surgical diagnosis	Cases: 29.7 (2.2) Controls: 31.1 (1.6)	Cases: 1235 (411.1) Controls: 1529 (474.4)	Stools; weekly from first stool up to NEC diagnosis or 5 weeks of life	Increased Cases: Proteobacteria (Enterobacteriaceae: Citrobacter koseri, Klebsiella pneumoniae, E. Coli); Bacteroidetes, Actinobacteria Decreased cases: Bradirhizobiaceae (Bacteroides), Lactobacillus sp	Increased Controls: Firmicutes (Lactobacillus)	Cases: lower microbial diversity with abnormal succession of microbial community	Cases: Male NI CS 7 (63.6%) ANS NI Intrapartum ATB 5 (45.5%) EBM 1 (9%) Time of onset 8 (5–13) d Controls: Male NI CS 24 (82.7%) ANS NI Intrapartum ATB 15 (51.7%) EBM 2 (6.9%)
Duan 2020 China [38]	Prospective; PCR-DGGE combined with DNA sequencing of V3 region of 16S rDNA	GA < 37 GW Cases: 28 (16 Bell’s I, 11 Bell’s II, 1 Bell’s III) Controls: 30 NEC Bell’s stage ≥ I	Cases: 34.2 (1.3) Controls: 34.7 (1.6)	Cases: 2.2 (0.4) Controls: 2.4 (0.5)	Stools; days 1, 3, 5, 7, 9 after admission (admission = diagnosis?), and at discharge	Increased Cases: Bacteroides and Klebsiella (higher number of samples) Decreased Cases: E. Coli, Bifidobacterium, Lactobacillus	Increased Controls: E. Coli, Bifidobacterium, Lactobacillus	Cases: lower Shannon’s diversity index (1.97 (0.54) vs. 2.68 (0.31)) gradually increasing over time; Species richness 7.68 (0.73) vs. 15.47 (2.62)	Cases: Male 15 (53.6%) CS 14 (50%) ANS, Antepartum ATB, Time of onset NI EBM 16 (57%) Controls: Male 14 (46.7%) CS 15 (50%) ANS, Antepartum ATB NI EBM 17 (56.7%)
Feng 2019 China [39]	Case-control; V3–V4 region of 16S rRNA gene sequencing	GA > 28 GW Cases: 16 Controls: 16 NEC Bell’s stage II or III	Cases: 34.8 (33.4–36.1) Controls: 35.1 (33.1–36.5)	Cases: 2325 (2063–2575) Controls: 2345 (2025–2675)	Stool; average of <10 h after NEC diagnosis; controls at the same postnatal day	No significant difference Increased Cases: Actinobacteria (Propionibacteriales) and Bacteroides Decreased Cases: Lactobacillus, Phascolarctobacterium and Str. salivarius	No significant difference Increased Controls: Lactobacillus, Phascolarctobacterium, Str. Salivarius Decreased Controls: Bacteroidetes	No difference in total diversity as expressed by Chao-i (*p* = 0.40)	Cases: Male 8 (50%) CS 8 (50%) ANS, Antepartum ATB NI EBM 3 (18.7%) Mean time of onset 9 (5–16) d Controls: Male 8 (50%) CS 8 (50%) ANS, Antepartum ATB NI EBM 5 (31.2%)
Heida 2016 Netherlands [40]	Case-control in prospective cohort trial; V3–V4 region of 16S rRNA gene analysis	≤30 GW, BW ≤ 1000 g or ≤32 GW and SGA with BW ≤ 1200 g, or with cardiovascular disease and reduced splanchnic blood flow, or neonates exposed to maternal indomethacin Cases: 11 NEC Controls: 22 NEC with pneumatosis intestinalis, or PVG, or both (Bell’s stage ≥II)	Cases: 27 (24–29) Controls: 26 (24–29)	Cases: 970 (560–1630) Controls: 995 (615–1735)	Stool; twice a week from meconium to last 2 faeces prior to NEC	Increased Cases: Cl. Perfrigens (8.4% and 6%) and Bacteroides dorei (0.9% and 0.7%) (meconium and 2 timepoints before NEC); Decreased Cases: Cl. Difficile (0.02%) (meconium), Staphylococci (0.5%) (1 timepoint before NEC);	Increased Controls: Cl. Difficile (2.7%) (meconium), and Staphylococci (23%) (1 timepoint before NEC) Decreased Controls: Cl. Perfrigens (0.1% and 0.003%) and Bacteroides (0.2% and 0.005%)	Microbial diversity (Simpson index) not associated with NEC development	Cases: Male 6 (54.5%) CS 4 (36.4%) ANS, EBM NI Peripartum ATB 3 (27%) Time of onset 12.5 (range 4–43) d Controls: Male 10 (45.5%) CS 13 (59%) ANS, EBM NI Peripartum ATB 5 (23%)
Itani 2018 Lebanon [41]	Case-control; quantitative PCR (qPCR); V3–V4 region of 16S rRNA analysis via TTGE;	GA < 36 GW (27–35 GW) Cases: 11 NEC Controls: 11 NEC Bell’s stage II or III	Cases: 31.2 (2.4) Controls: 31.4 (2.2)	Cases: 1516 (365) Controls: 1733 (481)	Stool; weekly from first meconium; Samples from NEC subjects divided into: before NEC diagnosis (TP1, >10 days before), at NEC diagnosis (TP2, <72 h from NEC), after NEC diagnosis (TP3, 7 to 10 days after);	qPCR: Increased cases: Staphylococci (*p* = 0.003); Decreased cases: lactobacilli (*p* = 0.048) and Enterococci (*p* = 0.039); TTGE: no clusterisation;	qPCR: Increased controls: Bifidobacterium and lactobacilli	TTGE: simple fecal microbiota in both groups (NMB 5.9, 1–10 vs. 6.7, 2–11)	Cases: Male 5 (45.5%) CS 11 (100%) ANS, Antepartum ATB NI EBM 4 (36.4%) Mean time of onset 22.6 (11.9, 10–50) d Controls: Male 8 (72.7%) CS 11 (100%) ANS, Antepartum ATB NI EBM 7 (63.6%)
Leach 2015 Australia [42]	Case-control; 16S rDNA gene analysis using next-generation sequencing techniques	GA 24–32 GW Cases: 4 NEC Controls: 18 NEC Bell’s stage II or III	Cases: 27.4 (2.5) Controls: 27.9 (0.7)	Cases: 1060 (346) Controls: 1204 (182.3)	Stools; first meconium, then daily for the first week, then weekly for the first 4 weeks; In cases, samples collected daily for the first week after diagnosis, then weekly until discharge;	Increased Cases: no consistency at the phyla level; Corynebacterium striatum (*p* = 0.01) and Morganella Morganii (*p* = 0.02), Ps. Aeruginosas, Corynebacterium amycolatum	Increased Controls: Proteobacteria	Species diversity prior to NEC equivalent or higher than controls of 27–32 GW (exact values NI)	Cases: Male 2 (50%) CS 3 (75%) ANS, Antepartum ATB NI EBM 2 (50%) Time of onset 34 (range 10–79) d Controls: Male 9 (50%) CS 10 (55.5%) ANS, Antepartum ATB NI EBM 10 (55.5%)
Lindberg 2018 CT, USA [43]	Prospective case-control; V4 of 16S rRNA gene sequencing	GA < 30 GW Cases: 5 NEC Controls: 5 NEC Bell’s stage II or III	Cases: 25.4 (24–27) Controls: 25 (24–27)	Cases: 695.8 (516–1026) Controls: 663.4 (485–860)	Stool; weekly from first meconium until discharge;	Increased Cases: Proteobacteria (Enterobacteriacee and Trabulsiella)	Increased Controls: Firmicutes (Veillonella and Enterococcus)	Number of OTUs and Simpson diversity index not different	Cases: Male 4 (80%) CS 4 (80%) ANS, Antepartum ATB NI Time of onset 38.5 (range 24–50) d EBM (M or D) 5 (100%) Controls: Male 4 (80%) CS 4 (80%) ANS, Antepartum ATB NI EBM (M or D) 4 (80%)
Liu 2019 China [44]	Prospective; V3-V4 regions of 16S rRNA gene sequencing and quantification with QuantiFluor-ST	GA < 33 GW BW > 950 g Cases: 4 NEC Controls: 17 NEC Bell’s stage II or III	Cases: 29 (29–30) Controls: 31 (28–33)	Cases: 1416.3 (773.4–2149.1) Controls: 1527.4 (1391.6–1663.1)	Stools; weekly, from meconium until death or discharge; Timepoints: TP1/EPP within 3 DOL; TP2/EPO from TP1 to 4 days before NEC; TP3/LPO from TP2 and onset; TP4/ED first third interval of disease span; TP5/MD; TP6/LD; TP7/PD from end to discharge	Increased Cases: Bacilli (TP1); Peptoclostridium; rapid surge in Enterococcus, Staphylococcus, Peptoclostridium and Streptococcus after disease onset Decreased Cases: TP1: Enterococcus (0.51%); TP1-TP4 decrease in Lactococcus (24.54 > 0.94%)	Increased Controls: Firmicutes (Veillonella), progressive increase in Klebsiella, Escherichia and Shigella Decreased Controls: TP1 to TP4: decrease in Lactobacillus, Pseudomonas and Enterococcus	Decreasing trend in microbiome richness (Sobs and Shannon-i) over time in both groups (NEC: *p* = 0.044, 3.14 > 0.58, *p* = 0.01; Controls: *p* < 0.01, 2.77 > 1.004, *p* = 0.04); Shannon-i significantly lower in NEC at TP4 (*p* = 0.025)	Cases: Male 1 (25%) CS 4 (100%) ANS, Antepartum ATB, EBM NI Time of onset 16 (11–19) d Controls: Male 8 (47%) CS 17 (100%) ANS, Antepartum ATB, EBM NI
Mai 2011 Florida, USA [45]	Prospective; DNA extraction via DGGE and V6–V8 region of 16S rRNA sequencing	GA ≤ 32 GW BW ≤ 1250 g Cases: 9 NEC Controls: 9 NEC Bell’s stage II or III	Cases: 26.2 (23–29) Controls: 27.5 (26–30)	Cases: 913.6 (570–1225) Controls: 1058 (652–1269)	Stools; weekly from meconium until discharge, and within 72 h from NEC	Increased Cases: Proteobacteria from 1 week up to <72 h from NEC (34% increase); y-Proteobacteria (related to Enterobacteriaceae) Decreased Cases: Actinobacteria and Bacteroidetes; decrease in Firmicutes from 1 week up to <72 h from NEC (32% decrease)	Increased Controls: Firmicutes, Bacteroides Decreased Controls: Proteobacteria	Total number of OUTs (estimated by Chao-i curves) not different	Cases: Male 5 (55.5%) CS 4 (44.4%) ANS, Antepartum ATB NI Time of onset 23.7 (range 5–41) d Controls: Male 4 (44.4%) CS 4 (44.4%) ANS, Antepartum ATB NI Predominant milk reported
McMurtry 2015 Louisiana USA [46]	Prospective; V1 to V3 region of 16S rRNA gene sequencing	GA ≤ 34 GW BW ≤ 1500 g Cases: 21 NEC Controls: 74 NEC Bell’s stage II or III, divided into mild, severe, lethal	Cases: 27.2 (2.8) Controls: 28.3 (2.5)	Cases: 1037 (397) Controls: 1111 (370)	Stools; 1–5 days prior to NEC symptoms’ onset, or on day of diagnosis; controls with matched specimens	Decreased Cases: Actinobacteria (1.24, *p* = 0.009) and Firmicutes (Clostridia, 5.76, *p* = 0.004); Veillonella (0.71, *p* = 0.007) and Streptococcus (0.96, *p* = 0.002))	Increased Controls: Actinobacteria (1.67) and Firmicutes (Clostridia 18.8), Veillonella (6.61, *p* = 0.007) and Streptococcus (4.32, *p* = 0.002)	Low bacterial diversity in cases (Chao-i *p* < 0.0001, Shannon-i *p* < 0.0002)	Cases: Male 11 (52.4%) CS 12 (57.1%) ANS 14 (66.7%) Antepartum ATB NI EBM 7 (33.3%) Time of onset (time of samples) 26.7 (14.9) days Controls: Male 38 (51.4%) CS 56 (75.7%) ANS 42 (56.7%) Antepartum ATB NI EBM 17 (30%)
Morrow 2013 Ohio USA [26]	Cross-sectional; V3 to V5 regions of 16S rRNA gene sequencing by 454 FLX Titanium	<29 w, <1200 g Cases: 11 (8 Bell’s stage II and 3 Bell’s stage III) Controls: 21 NEC II or III Bell’s modified; Subtypes of NEC based on ordination of day 4–9 samples: -NEC-I (*n* = 4) dominated by Firmicutes (Bacilli) -NEC-II (*n* = 5) dominated by Proteobacteria (Enterobacteriaceae)	Cases: 25.5 (1.8) Controls: 25.9 (1.5)	Cases: 791 (212) Controls: 839 (187)	Stools; TP1 4–9 days; TP2 10–16 days	Increased Cases: TP1: Firmicutes (4/9, NEC-I: Enterococcus and Staphylococcus 98%), -TP2: Proteobacteria (5/9, NEC-II, Enterobacteriaceae) PPV of Firmicutes or Proteobacteria 88% Decreased Cases: Firmicutes and Actinobacteria, Propionibacterium (TP1)	Increased Controls: Propionibacterium (10/18, 56%), 80% Proteobacteria (12/18: Enterobacter, Escherichia), 20% Firmicutes (Enterococcus, Staphylococcus); Decreased Controls: TP1: Firmicutes (2/18: Enterococcus 62% and Staphylococcus 73%);	T1: Chao-i: Cases 9.2, Controls 18.4, *p* = 0.086; similar Simpson index (*p* = 0.221)	Cases: Male 6 (54.6%) CS 7 (63.6%) ANS 10 (90.9%) Antepartum ATB 6 (54.6%) EBM (M or D) 11 (100%) Time of onset: NEC-I 7–21 d, NEC-II 19–39 d Controls: Male 8 (38.1%) CS 14 (66.7%) ANS 19 (90.5%) Antepartum ATB 10 (47.6%) EBM (M or D) 11 (100%)
Normann 2013 Sweden [47]	Prospective case-control; V3-V4 regions of the 16S rRNA gene amplification and barcoded pyrosequencing	<28 GW Cases: 10 NEC Controls: 10 (+6 in sub-analysis) NEC Bell’s stage II or III with radiological pneumatosis and/or portal venous gas or intraoperative histopathology confirmation	Cases: 23.5 (22–25.5) Controls: 23.5 (22–25.6)	Cases: 582 (487–965) Controls: 570 (440–892)	Stools; weekly for first 7 weeks or until NEC diagnosis	Increased Cases: Bacillales and Enterobacteriaceae in the 1st week; Enterobacteriaceae in the 2nd week	Increased Controls: Firmicutes (Enterococcus and Bacilli)	No difference in Shannon-i between cases and controls	Cases: Male NI CS 4 (40%) ANS NI Antepartum ATB 9 (90%) EBM (M or D) 100% Time of onset 5–48 daysI Controls: Male NI CS 4 (40%) ANS NI Antepartum ATB 8/16 (50%) EBM (M or D) 100%
Olm 2019 CA USA [48]	Prospective; Extensive computational analyses to recover genome de novo, phylogeny, metabolic potential and replication rates;	Criteria NI Cases: 34 NEC Controls: 126 NEC NI	Cases: 28 (2.5) Controls: 29 (2.2)	Cases: 1154.5 (465.3) Controls: 1217 (388.5)	Stools; mostly first month of life (average 7.2 samples per infant), with focus on those immediately before NEC onset (<2 days before, “pre-NEC”)	Increased Cases: Enterobacteriaceae (*p* = 8.9 × 10^−7^) and Bacteroidetes after NEC development; K. pneumoniae 52% pre-NEC (*p* = 0.008) Decreased Cases: Firmicutes (*p* = 3.7 × 10^−7^) after NEC	Increased controls: Firmicutes Decreased Controls: Enterobacteriaceae (K. pneumoniae 23%)	NI	Cases: Male 15 (44.1%) CS 25 (73.5%) ANS, Antepartum ATB NI EBM 14 (41.2%) Mean time of onset 9 (9.8) d Controls: Male 61 (48.4%) CS 93 (73.8%) ANS, Antepartum ATB NI EBM 35 (27.8%)
Rozé 2017 France Clinical data from EPIPAGE 2; Microbial data from EPIFLORE [49]	Prospective; V3–V4 region of 16S rRNA gene pyrosequencing	24–31 GW, >7 DOL Cases: 106 NEC Controls: 3055 NEC Bell’s stage II or III	NI (divided into 3 subclasses)	Data for whole cohort Cases: NI Controls: 1175 (348)	Stools; at 7 and 28 DOL, at discharge (controls), at NEC diagnosis (cases); Culture-independent microbiota analysis on 15 cases and 57 controls	Increased Cases: Cl. sensu stricto (20.6%, *p* = 0.08) (neonatale and butyricum); Gammaproteobacteria (Enterobacteriaceae) Decreased Cases: Gammaproteobacteria (Klebsiella), *Staphylococcus* sp., *Enterococcus faecalis*	Increased Controls: *Staphylococcus* sp., *Enterococcus faecalis* Decreased Controls: Cl. sensu stricto (11.7%)	NI	Data for whole cohort: Cases: Male 63 (59.4%) CS, ANS, EBM, Antepartum ATB NI Median time of onset 26 (IQR 20–42) d Controls: Male 1548 (52.3%) CS, ANS, EBM, Antepartum ATB NI
Sim 2014 UK [50]	Prospective; V3-V5 regions of 16S rRNA gene sequen cing	<32 GW Cases: 12 NEC Bell’s stage II or III (analysed), 8 suspected NEC Controls: 44 (36 analysed) NEC according to VON criteria and staged according to Bell	Cases: 27 (25.7–28.4) Controls: 27.5 (25.4–29)	Cases: 845.4 (685–898.8) Controls: 1005.9 (755–1239.5)	Stools, every sample from recruitment until discharge	Increased Cases: Clostridia (Confirmed NEC, *p* = 0.006)	Increased Controls: Klebsiella, Staphylococcus, Enterobacteriaceae, Enterococcus, and Bifidobacterium	NI	Cases: Male 5 (41.7%) CS 7 (58.3%) ANS, EBM NI Peripartum ATB 2 (16.7%) Time of onset 27.5 (IQR 20.8–37.5) days Controls: Male 17 (47.2%) CS 18 (50%) ANS, EBM NI Peripartum ATB 11 (30.6%)
Stewart 2012 UK [51]	Prospective; V3 region of 16S rRNA gene amplification and profiling via DGGE	Preterm infants (criteria NI) Cases: 8 NEC Controls: 22 NEC categorized by 2 neonatologists as medical (pneumatosis, no surgery) or surgical	Cases: 25.7 (1.7) Controls: 27.2 (2.3)	Cases: 842.5 (227.4) Controls; 1027 (338.4)	Stools; weekly from meconium 7 NEC cases with molecular data	Increased Cases: *Staphylococcus* spp. (CONS 45%), Enterobacter spp. Decreased Cases: Ent. Faecalis (31%)	Increased Controls: *Enterococcus* spp. (Ent. Faecalis 57%) and Streptococcus spp. Decreased Controls: CONS (30%)	Low bacterial diversity increasing over time for whole cohort	Cases: Male 7 (87.5%) CS 4 (50%) ANS, Antepartum ATB, EBM NI Time of onset 16.7 (4.8) d Controls: Male 13 (59%) CS 11 (50%) ANS, Antepartum ATB, EBM NI
Stewart 2013 UK [52]	Prospective; PCR-DGGE analysis of targeted V3 region of 16S rRNA gene and pyrosequencing	GA < 32 GW from multiple birth Total cohort: 27 (12 twin pairs and 1 triple set), 5 of which developed NEC NEC confirmed by two neonatologists (definition NI)	Cases: 27.6 Controls: 27.2	Cases: 1106 Controls: 975.2	Stools, from birth to discharge	Increased Cases: Escherichia sp.	-	Reduced diversity in cases	Cases: Male 4 CS 3 ANS, Antepartum ATB, EBM NI Time of onset 25.4 (range 16–45) d Controls: Male 16 CS 17 ANS, Antepartum ATB, EBM NI
Stewart 2016 UK [29]	Prospective; V4 region of 16S rRNA gene sequencing	<32 GW Cases: 7 Controls: 28 NEC “defined rigorously” by one senior clinician and two senior research clinicians, and classified as either surgical or medical, where pneumatosis was required for medical cases	Cases: 26 (23–30) Controls: 27 (24–30)	Cases: 760 (500–1470) Controls: 910 (545–1810)	Stools; (DOL) −14 (TP1), −7 (TP2), 0 (TP3), +7 (TP4), and +14 (TP5) “Pre-NEC” > 10 days from onset	No clear causative organism diagnostic for NEC; No PGCTs assigned to preNEC samples PGCT 2 (Klebsiella and Enterococcus) and PGCT 5 (Escherichia) most associated with preNEC samples; Decreased Cases: PGCT 6 (Bifidobacterium)	Increased Controls: PGCT 6 (Bifidobacterium predominance)	Controls: higher alpha diversity and Shannon diversity (PGCT 6) compared to Cases; progressive lower acquisition of diversity	Cases: Male 3 (42.9%) CS 3 (42.9%) ANS, Antepartum ATB, EBM NI Time of onset 26.4 (14–42) days; Controls: Male 20 (71.4%) CS 15 (53.6%) ANS, Antepartum ATB, EBM NI
Torrazza 2013 Florida USA [53]	Prospective; V6–V8 region of 16S rRNA gene analysed by DGGE and PCR amplification, sequencing	≤32 GW Cases: 18 NEC Controls: 35 NEC with clinical and radiologic signs or necrotic bowel at surgery	Cases: 27.4 (2.6) Controls: 28.5 (2.2)	Cases: 1073 (394) Controls: 1246 (350)	Stools; from birth (meconium) until discharge, analysed: TP1 2 weeks prior to NEC, TP2 1 week prior to NEC, TP3 closest to NEC diagnosis; matched for controls;	Increased Cases: -TP1: Proteobacteria (61%, *p* < 0.001)) -TP2: Actinobacteria 3% and Proteobacteria (*p* < 0.001) -TP3: Firmicutes 72%; Klebsiella granulomatis, Klebsiella pneumoniae and Clostridium perfringens, St. epidermidis Lower Cases: -TP1: Bacteroidetes	Decreased Controls: -TP1: Proteobacteria (19%) -TP2: Actinobacteria 0.4%	Similar Chao-i (species richness as alpha diversity) at all TPs; Significant different beta diversity (UNIFRA metric) at TP1 (*p* < 0.05) indicating a similar total number of species but different bacteria and proportions between cases and controls	Cases: Male 12 (66.7%) CS 9 (50%) ANS 11 (61%) Antepartum ATB 13 (72.2%) EBM 27.8% Time of onset 17.8 (12.8) Controls: Male 17 (48.6%) CS 23 (65.7%) ANS 20 (57%) Antepartum ATB 29 (82.9%) EBM 57.1%
Wandro 2018 California USA [32]	Retrospective; 16S rRNA gene sequencing	VLBW (<1500 g) Cases: 3 NEC Controls: 21 NEC definition NI	Cases: 25.6 Controls: 27.4	Cases: 920 Controls: 1018.6	Stools; over first 6 weeks of life (variable timepoints, between 7 and 75 DOL)	No single bacterial OUT or community composition consistent of NEC or LOS		Lower bacterial abundances in infants developing NEC or LOS (*p* < 0.001); Alpha diversity (Shannon-i) increasing overall with age	Cases: Male NI CS 1 (33.3%) ANS, Antepartum ATB NI EBM 1 (33.3%) Mean time of onset 33 days Controls: Male NI CS 16 (76.2%) ANS, Antepartum ATB NI EBM 12 (57.1%)
Wang 2010 IL, USA [54]	Prospective case-control; 16S rRNA gene sequencing	25–32 GW Cases: 10 NEC Controls: 10 NEC Bell’s stage II or III	Cases: 25–32 Controls: 26–32	Birth weight NI	Stools; at NEC diagnosis (<1 day) (range 4–49 days)	Increased Cases: Proteobacteria (90.7% RA; *p* = 0.001; Gammaproteobacteria); at OTU level: *Klebsiella pneumoniae*, *Shigella dysenteriae*, *Enterobacter hormaechei* and *Escherichia coli.* Decreased Cases: Firmicutes (9.1%, *p* = 0.001)	Increased Controls: Firmicutes (57.8%), Bacteroidetes (2.4%), Fusobacteria (0.5%); at OTU level: *Veillonella* sp., *Escherichia coli*, *Enterococcus* sp., *Staphylococcus* sp., *Enterobacter aerogenes (all 90%)* Decreased Controls: Proteobacteria (34.9% RA, Gammaproteobacteria)	Low diversity in preterm infants, especially those with NEC: Shannon-i by T-RFLP cases 1.13 vs. controls 1.88, *p* = 0.035; OTUs cases 10.4 (6.1) vs. controls 19 (6.7), *p* = 0.008; Shannon-i by library cloning csses 1.19 (0.62) vs. controls 1.99 (0.55), *p* = 0.005	Cases: Male 6 (60%) CS 8 (80%) ANS, Antepartum ATB NI EBM 4 (40%) Mean time of onset 5–49 days Controls: Male 7 (70%) CS 9 (90%) ANS, Antepartum ATB NI EBM 6 (60%)
Ward 2016 OH, USA [55]	Prospective caso-control; Shotgun metagenomic sequence analysis and pangenome-based computational analysis for E. Coli-specific gene content	Cases: 27 NEC (<30 GW) Controls 1: 117 (<30 GW) Controls 2: 22 (>37 GW) NEC Bell’s stage II or III (SIP excluded)	Cases: 26 (23–28) Controls 1: 26 (23–29) Controls 2: 39 (38–41)	Cases: 850 (415–1340) Controls 1: 904 (520–1741) Controls 2: 3476 (2217–4173)	Stools; between 3 and 22 DOL: TP1 3–9 DOL, TP2 10–16 DOL, TP3 17–22 DOL;	TP1 (8 NEC cases): taxa similar to preterm without NEC (Firmicutes Bacilli with S. epidermidis, Lactobacillales with E. fecalis, Gammaproteobacteria with Enterobacter) TP2 (15 NEC cases): not different from controls 1 (S. epidermidis, E. faecalis, E. cloacum, S. marcescens) TP3 (7 NEC cases): E.faecalis and Streptococcus; E. Coli; decreased Veillonella	TP1: controls 2: Actinobacteria (Bifidobacterium spp) and Bacteroidetes, Firmicutes, Negativicutes (Veillonella) TP2: controls 2 increased in all taxa except S. epidermidis, E. faecalis, E. cloacum, S.marcescens TP3: controls 2: all taxa except E.faecalis and Streptococcus	TP1: similar Shannon-i between cases and controls 1 (1.1 (0.79) vs. 0.96 (0.56), *p* =0.59) TP2: similar Shannon-i between cases and controls 1 (1.01 (0.92) vs. 1.15 (0.72), *p* = 0.52) TP3: Cases with less diversity than controls 1, but not significantly (SI 0.87 (0.63) vs. 1.32 (0.68), *p* =0.12)	Cases: Male 15 (56%) CS 16 (59%) ANS NI Peripartum ATB 12 (44%) Human milk ≥ 75% in first month 17 (63%) Mean time of onset 21 (7.4) days Controls 1: Male 61 (52%) CS 70 (60%) ANS NI Peripartum ATB 79 (68%) Human milk ≥ 75% in first month 86 (74%) Controls 2: Male 11 (50%) CS 10 (45%)
Warner 2016 MO USA [56]	Prospective; V3 to V5 regions of 16S rRNA genes pyrosequencing	VLBWI ≤ 1500 g, >7 days of life; Cohort 1: Cases: 28 NEC Controls: 94 Cohort 2: Cases: 18 NEC Controls: 26 NEC Bell’s stage II or III (SIP and CHD excluded)	Cohort 1: Cases: 26 (24.7–27.9) Controls: 27 (25.9–28.7)	Cohort 1: Cases: 795 (720–980) Controls: (940 (800–1500)	Stools; up to and including the day before NEC or at 60 days of age (whichever came first), divided into TP of 15 days	Increased Cases: Gammaproteobacteria (*p* = 0.001) (E. Coli, Enterobacter, Klebsiella) Decreased Cases: Anaerobic bacteria (Negativicutes, *p* = 0.0013; Clostridia-Negativicutes, *p* = 0.005)	Increased Controls: Negativicutes and Clostridia-Negativicutes	Shannon-i increasing in stools from controls, not from cases, with significantly discordant trend (*p* = 0.0004)	Cases: Male 18 (64%) CS 20 (71%) (75%) ANS, Antepartum ATB NI Exposure to human milk >50% 21 (75%) Time of onset 24 (19–48) days Controls: Male 45 (48%) CS 72 (77%) (53%) ANS, Antepartum ATB NI Exposure to human milk >50% 50 (53%)
Zhou 2015 MO USA [57]	Prospective case-control; V3–V5 region of 16S rRNA gene sequencing	<32 GW Cases: 12 NEC (6 medical, 6 surgical) Controls: 26 NEC Bell’s stage II or III (SIP excluded)	Cases: 27.8 (24–31) Controls: 27.9 (24–31)	Cases: 1048 (940–1860) Controls: 1092 (520–1800)	Stools, from birth to discharge or 60 DOL (median sampling interval of 3 days)	Increased Cases: Clostridia, in particular Cl. Sensu stricto (early onset NEC, <22 DOL); Gammaproteobacteria (Pseudomonas, Pasteurella,m Serratia, Klebsiella) and Escherichia, Shigella, Cronobacter, (Late onset NEC, >22 DOL) More differences at 2 weeks of life	Increased Controls: Veillonella (1–3 days prior to NEC, *p* = 0.005) Decreased Controls: Pasteurella	Increased richness in controls (*p* = 0.03); progressive increase in richness and Shannon-i over 2 months of life in cases (*p*<0.05)	Cases: Male 7 (58%) CS 9 (75%) ANS, EBM NI Antepartum ATB 4 (33%) Time of onset 25.5 (IQR 16.8–37) days Controls: Male 14 (54%) CS 17 (65%) ANS, EBM NI Antepartum ATB 6 (23%)

Abbreviations: ANS = antenatal steroids; CAM = chorioamnionitis; CCC = complex congenital cardiopathy; DGGE = denaturing gradient gel electrophoresis; EBM = exclusive breast milk; F = French; GA = gestational age; GW = gestational weeks; g = grams; h = hours; MP = multiple pregnancy; NEC = necrotizing enterocolitis; NI = no information; NMB = number of major bands; OUT = Operational Taxonomic Unit; PGCT = preterm gut community type; PVG = portal venous gas; RA = relative abundance; RCT = randomized controlled trial; SIP = spontaneous intestinal perforation; TTGE = temperature temporal gel electrophoresis; TP = timepoint; VLBW = very low birth weight; w = weeks.

**Table 3 nutrients-14-03859-t003:** Articles not eligible for qualitative analysis and reasons for exclusion.

Study	Reason for Exclusion
Abdulkadir et al. 2016 [58]	Missing data
Barron et al. 2017 [59]	Missing data
De Magistris et al. 2015 [60]	Full text unavailable
Feng et al. 2017 [61]	Full text unavailable
Itani et al. 2019 [62]	Same population of Itani et al. 2018 [40]; brief report
Raveh-Sadka et al. 2015 [63]	Missing data; no comparison between NEC and Controls
Romano-Keeler et al. 2018 [64]	Comparison of NEC and surgical patients; no comparison with healthy controls
Sinclair et al. 2020 [65]	Targeted metabolome analysis
Smith et al. 2011 [66]	Analysis of inflamed intestinal tissue; no controls
Stewart et al. 2019 [67]	Comparison between NEC and SIP, no controls
Yang et al. 2015 [68]	Comparison between NEC and congenital intestinal atresia, no controls

## Data Availability

Data is contained within the article or Appendix A.

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
