# Peer review of "The Metabolome and the Gut Microbiota for the Prediction of Necrotizing Enterocolitis and Spontaneous Intestinal Perforation: A Systematic Review"

_nutrients, 2022, doi:10.3390/nu14183859_

Round 1
Reviewer 1 Report
Drs Moschino and colleagues provide a literature review on the role of metabolomic signatures and microbiome data in predicting necrotizing enterocolitis in preterm infants.
The topic is relevant as NEC continues to occur despite efforts to reduce this condition.
Introduction:
-Sufficient information is provided and relevant aspects are discussed.
Methods:
-methodological aspects are explained in detail
-as the search has identified SIP as the main differential diagnosis this could be included in the title, and the fact that there is no data available in cases of SIP is a relevant result.
-results are presented in detail and supported
discussion:
-important limitations are discussed
-the current limitations in understanding of the pathogenesis of NEC are outlined and suggestions are made for a prospective characterization approach
-maybe fecal volatile organic compounds could be discussed as well .
Queries and suggestions:
L 116: studies being excluded due to valid reasons ? does this mean they were excluded based on the prespecified criteria? Or for reasons of validity?
L 154: This appears to be a repetition in the same sentence- linoleate metabolism.
L 187: ...maybe: were the techniques used most frequently...
L 195: cases were characterized by a significantly lower abundance
references seem to be duplicated -ref 11 Garg et al is the same as ref 73.
Author Response
We thank the reviewer for his/her helpful remarks to ameliorate our manuscript.
We agree that the topic is relevant and we have highlighted this aspect in the Introduction and the Discussion.
- We have changed the title to include “spontaneous intestinal perforation”, as it goes in the main differential diagnosis of NEC
- We have added “Importantly” in line 30 of the abstract to highlight the fact that no data are available in cases of SIP
- Fecal VOCs: we have mentioned this technique in the revised version of the manuscript in the Discussion section (L523-528) and we have included 3 references regarding this (ref. 109-111).
- L 116-117: We have specified the reasons for articles’ exclusion, as reported in the PRISMA flow diagram (Figure 1).
- L 157: we have removed the repetition “linoleate metabolism”
- L 190: We have changed the sentence: “The sequencing of … was the technique used most frequently”.
- L 200: The sentence has been corrected: “.. Cases were characterized by a significant lower relative abundance …”.
- We have deleted reference 73, as it was indeed a duplicate of ref. 11 (now ref. 12).
Reviewer 2 Report
Necrotizing enterocolitis (NEC) is a life-threatening disease affecting mostly very low birth weight (VLBW) infants. The mortality rate among extremely low birth weight (ELBW) infants is about 50%. Infants at 22-23 weeks are at greater risk of sepsis and NEC compared with more mature infants, even those at 24-25 weeks of gestation. Several hospitals worldwide report >50% survival rate as early as 22 weeks of gestation, but data on long-term outcomes are hardly available.
https://www.jpeds.com/article/S0022-3476(21)00218-3/fulltext
In this paper, NEC was defined as Bell stage II or greater. Spontaneous intestinal perforation (SIP) was diagnosed if free air in the abdomen for a reason besides NEC. The denominator includes all live births (n = 943 at 22 weeks; n = 2712 at 23 weeks; n = 3764 at 24 weeks; n = 4366 at 25 weeks; n = 5278 at 26 weeks; n = 6161 at 27 weeks).
Rates of invasive fungal infection are also much higher in this population. Measures such as human milk feeding, probiotics, and attention to meconium passage and feeding tolerance may mitigate risks of adverse gastrointestinal sequelae.
NEC may be related to several complications (short bowel syndrome, intestinal failure, neurodevelopmental delay), in recent years targeted research on biomarkers for early prediction and diagnosis of NEC has been implemented including the “omics” technologies.
In a systematic review, Laura Moschino et al provide an updated perspective of the literature in which untargeted metabolomics and gut microbiota analysis has been applied for the prediction and diagnosis of NEC (The metabolome and the gut microbiota for the prediction of necrotizing enterocolitis: a systematic review).
The authors detected 1191 studies, 1158 were excluded due to valid reasons.
Six studies applied untargeted metabolomic analysis, used untargeted microbiota analytic techniques, and 4 applied both. Only five studies applying metabolomics and 20 studies evaluating gut microbiomes were identified.
As highlighted by the authors, most included studies have a small sample size, the inclusion criteria are variable, and different techniques were applied, none of the included studies compared NEC versus SIP, which is the most common surgical disease in the differential diagnosis of NEC.
Meconium microbial composition, the in-utero environment varies depending on gestational age, with the potential involvement in the subsequent development of sepsis and NEC; however, NEC does not occur in utero.
https://www.mdpi.com/2075-1729/11/2/148
Neonatal microbiota seems to be influenced by the mode of delivery, and drug administration. and early neonatal nutrition. Breastfeeding allows the transmission of a unique lactobiome, able to modulate and positively affect the neonatal gut microbiota.
Mode of delivery is a key factor determining early microbial colonization. Newborns born by vaginal delivery acquire microbial communities similar to the maternal gut and vagina; on the contrary, infants born by cesarean section acquire environment-like bacteria, such as Staphylococcus spp., Corynebacterium spp., and Propionibacterium spp.
Metagenomic analysis of human milk by total DNA reported that human milk contains >360 prokaryotic genera, with Proteobacteria (65%) and Firmicutes (34%) as the predominant phyla, and with Pseudomonas spp. (61.1%), Staphylococcus spp. (33.4%), and Streptococcus spp. (0.5%) as the predominant genera.
https://bmcmicrobiol.biomedcentral.com/articles/10.1186/1471-2180-13-116
A significant change in the composition of the breast milk microbiota has been observed over the lactation stage. The most common genera in colostrum samples detected by 16S sequencing included Leuconostoc spp., Weissella spp., Staphylococcus spp., Streptococcus spp., and Lactococcus spp. according to analyses of breast milk samples from 18 mothers from Finland.
In addition, several yeasts and fungi have been identified in breast milk samples collected from healthy mothers.
Pannaraj P.S. et al. reported that during the first month of life, infants who obtained through breastfeeding 75% or more of their daily milk intake, received a mean of 27.7% of the bacteria from breast milk and 10.3% from areolar skin. Bacterial and composition diversity depended on the proportion of daily breast milk intake.
https://pubmed.ncbi.nlm.nih.gov/28492938/
A significant change in the composition of breast milk microbiota was observed over the lactation stage. The most common genera in colostrum samples detected by 16S sequencing included Leuconostoc spp., Weissella spp., Staphylococcus spp., Streptococcus spp., and Lactococcus spp. according to analyses of breast milk samples from 18 mothers from Finland. From 1 to 6 months after delivery, a significant increase was observed in Veillonella spp., Prevotella spp., Bifidobacterium spp., Enterococcus spp. and Leptotrichia spp.
Drago et al. analyzed the microbiota network of colostrum and mature milk of Italian and Burundian mothers and observed that all samples showed different bacterial distributions in the microbiota network.
https://europepmc.org/article/med/27983720
In the present systematic review Laura Moschino et al concluded that in studies applying untargeted gut microbiota analysis, the sequencing of the V3-V4 21 or V3 to V5 regions of the 16S rRNA was the most used technique. At the phylum level, NEC specimens were characterized by an increased relative abundance of Proteobacteria compared to controls. At the genus level, pre-NEC samples were characterized by lack or decreased abundance of Bifidobacterium.
At the species level Bacteroides dorei, Clostridium perfringens, and perfringens-like strains dominated early NEC specimens. Upon diagnosis of NEC Clostridium butyricum, Clostridium neonatale and Propionibacterium acnes were detected. Six studies found a lower Shannon diversity index in NEC pathogens compared to controls.
They proposed a Flow chart to conduct future studies applying untargeted metabolomics and microbiota analysis to explore biomarkers to predict NEC.
In conclusion, this systematic review article summarized the available data with regard to the early detection of NEC at metabolome and gut microbiota levels.
Besides the limitations (listed also by the authors) this paper is well-written, and carefully structured, listing more than 100 references. I suggest including some of the above-mentioned literature data to provide a deeper insight into the complexity of diagnostic challenges.
Author Response
We thank the reviewer for his/her helpful remarks to ameliorate our manuscript.
We have included the following references in the revised manuscript, in order to provide a deeper insight into the complexity of the diagnostic challenges for NEC, as suggested by the reviewer.
These are:
- Ref. 3 in the Introduction
- Ref. 88, 89, 90 in the Discussion; L 413-416 have been added as well.